# Diethyl Blechnic Exhibits Anti-Inflammatory and Antioxidative Activity via the TLR4/MyD88 Signaling Pathway in LPS-Stimulated RAW264.7 Cells

**DOI:** 10.3390/molecules24244502

**Published:** 2019-12-09

**Authors:** Jia He, Shan Han, Xin-Xing Li, Qin-Qin Wang, Yushun Cui, Yangling Chen, Hongwei Gao, Liting Huang, Shilin Yang

**Affiliations:** 1College of Pharmacy, Guangxi University of Chinese Medicine, Nanning 530000, China; hj1210026043@sina.com (J.H.); hanshan201807@163.com (S.H.); kini9601@foxmail.com (X.-X.L.); 18329759158@163.com (Q.-Q.W.); 13557890205_cyl@sina.com (Y.C.); yangshilin@suda.edu.cn (S.Y.); 2Guangxi Engineering Technology Research Center of Advantage Chinese Patent Drug and Ethnic Drug Development, Nanning 530200, China; 3State Key Laboratory of Innovative Drug and Efficient Energy-Saving Pharmaceutical Equipment, Jiangxi University of Traditional Chinese Medicine, Nanchang 330004, China; cuiys2013@163.com

**Keywords:** diethyl blechnic, inflammation, TLR4/MyD88, NF-κB, oxidative stress

## Abstract

Inflammation is a common pathogenesis in many diseases. *Salvia miltiorrhiza* Bunge (Danshen), a traditional Chinese medicine, has been considered to have good anti-inflammatory effects. In the present study, we investigated the anti-inflammatory effect of diethyl blechnic (DB), a novel compound isolated from Danshen, and its possible mechanisms in lipopolysaccharide (LPS)-induced RAW264.7 macrophages. The results showed that DB can inhibit the LPS-induced pro-inflammatory cytokines release of prostaglandin E2 (PGE2) and mRNA expression of TNF-α, IL-6, and IL-1β. In addition, the results of the flow cytometry assay and the fluorometric intracellular ROS kit assay indicated that DB reduced the generation of ROS in LPS-stimualted RAW264.7 cells. DB reversed the LPS-induced loss of the mitochondrial membrane potential (MMP). Furthermore, DB suppressed the LPS-stimulated increased expression of Toll-like receptor 4 (TLR4), myeloid differential protein-88 (MyD88) and phosphorylation of TAK1, PI3K, and AKT. DB promoted NF-E2-related factor 2 (Nrf2) into the nucleus, increased the expression of heme oxygenase-1 (HO-1) and NAD(P)H dehydrogenase [quinone] 1 (NQO1) and reduced the expression of Keap1. In summary, DB may inhibit LPS-induced inflammation, which mainly occurs through TLR4/MyD88 and oxidative stress signaling pathways in RAW264.7 cells.

## 1. Introduction

Inflammation is the body’s physiological defense response to different stimuli [1,2]. Moderate inflammation can enhance the body’s autoimmune response, but excessive inflammation activates immune cells, which can damage both healthy and damaged tissues. Stimulated cells proliferate, differentiate, and then secrete inflammatory factors, chemokines, and other factors to promote the occurrence and development of tissue inflammation [3]. Macrophages have an important role in an inflammatory response and are the central cells that initiate the production of inflammatory mediators [4]. LPS is an important inflammation trigger, which induces cytokines to participate in inflammation [5]. So far, the LPS-induced RAW264.7 model has been widely used in the study of inflammation [6].

Screening of anti-inflammatory drugs based on key molecules associated with the inflammatory signaling pathway has become a hot research topic over recent years. NF-κB is a multipotent transcription factor, which participates in transcriptional regulation of various inflammatory mediators [7], including pro-inflammatory cytokines (TNF-a, IL-6, IL-1β), chemokines and immune receptors [8]. The role of TLRs in the upstream signaling molecules of inflammation and related diseases is highly implicated [9]. When induced by LPS, MyD88 is recruited to TLR4, which then activates phosphatidylinositol 3-kinase (PI3K), protein kinase B (AKT), and transforms growth factor-b-activated kinase 1 (TAK1). This process induces metastasis of NF-κB heterodimer into the nucleus, initiates the transcription of inflammation target genes, and then amplifies the inflammation signal [10,11,12]. This data suggest that the inhibition of TLR4/MyD88/PI3K and NF-κB signaling pathways could be an effective management strategy for a number of inflammation diseases.

Keap1/Nrf2/HO-1 is a classic antioxidant pathway [13], which can negatively regulate inflammatory regulators and enzymes, inhibit oxidative stress and overexpression of pro-inflammatory factors [14]. Under normal physiological conditions, NRF2 binds to Keap-1 and is then inhibited. Once oxygen free radicals are produced, the expression of Keap-1 decreases and the free Nrf2 increases. At the same time, the free Nrf2 translocates into the nucleus and binds to the antioxidant elements of antioxidant genes such as NQO1, HO-1, which initiate compensatory expression of genes and eliminate oxygen free radicals and oxidative stress through the biological activities of NQO1 and HO-1 [15]. Therefore, the nuclear translocation of NRF2 and the increased expression of HO-1 can effectively inhibit inflammation reaction.

Natural products have been reported as an important source for anti-inflammatory agent discovery [16]. Our previous study has shown that diethyl blechnic (DB), a novel compound isolated from Danshen, has significant anti-inflammatory properties and the MAPKs pathway and NF-κB pathways participated in its anti-inflammatory process [17,18], yet, its antioxidative effect has not yet been explored. In this study, following the previous study, we continued to examine the protective effect of DB and its molecular mechanisms against LPS-induced inflammation and oxidative stress in vitro.

## 2. Results

### 2.1. DB Suppresses LPS-Induced inflammatory Response in RAW264.7 Cells

Diethyl blechnic (Figure 1A), a novel natural product isolated from *Salvia miltiorrhiza* Bunge, showed no significant cytotoxicity in RAW264.7 cells compared with normal control (Figure 1B). In addition, DB suppressed the LPS-induced inflammatory by inhibiting nitrite level (Figure 1C), iNOS and COX-2 expression (Figure 1D), and NO production (Figure 1E,F) in LPS-induced RAW264.7 cells, which is consistent with our previous study [18].

### 2.2. DB Suppresses the Release of Pro-Inflammatory Cytokines in LPS-Stimulated RAW264.7 Cells

Next, we investigated the effect of DB on the release of pro-inflammatory cytokines in LPS-induced RAW264.7 cells. The results indicated that DB decreased the levels of pro-inflammatory cytokines PGE2 (Figure 2A), TNF-α and IL-6 (Appendix AA,B). In addition, the LPS-increased mRNA expression levels of IL-1β, IL-6 and TNF-α were significantly reversed following DB pretreatment (Figure 2B). These data suggest that DB may significantly decrease the release and the gene expression of pro-inflammatory cytokines.

### 2.3. DB Suppresses LPS-Induced NF-κB Nuclear Translocation in RAW264.7 Cells

NF-κB is a group of nucleoprotein factors that regulate the expression of a wide range of genes, which have a pivotal role in LPS-induced inflammatory processes [19,20]. Some studies have reported that LPS may induce the translocation of NF-κB/p65 from the cytoplasm to the nucleus, and then regulate the release of large amounts of inflammatory mediators such as TNF-α, IL-6, IL-1β, NO, and iNOS [20]. In this study, we found that LPS was able to increase the mRNA expression of NF-κB/P65 and IκBα in RAW264.7 cells, which was suppressed by DB (Figure 3A,B). 

In addition, LPS decreased the expression of NF-κB in the cytoplasm and increased expression of NF-κB in the nucleus, which was partially reversed by DB (Figure 3C). Furthermore, in LPS-stimulated RAW264.7 cells, immunofluorescence analysis indicated that DB suppressed the translocation of NF-κB/p65 into the nucleus (Figure 3D).

### 2.4. DB Suppresses the TLR4-MyD88 Pathway in LPS-Stimulated RAW264.7 Cells

TLR4, a transmembrane receptor located on the surface of many cells, has a pivotal role in inflammatory processes. The stimulation of LPS causes TLR4 to form a dimer, which then modulates the NF-κB signaling pathway, thereby spawning a pathogen-specific innate immune response through the release of pro-inflammatory cytokines [21]. To further study the specific internal mechanisms of DB, the expression of TLR4 and MyD88, phosphorylation of TAK1, PI3K, and AKT were analyzed in LPS-induced RAW264.7 cells. The results showed that LPS treatment activated the TLR4/MyD88 pathway, which was significantly suppressed by DB (Figure 4).

### 2.5. DB Suppresses the Loss of MMP and LPS-Induced ROS Generation

Mitochondrial membrane potential (MMP) is formed by the asymmetric distribution of protons and other ions on both sides of the intima during respiratory oxidation and is important for maintaining the normal physiological function of cells [22,23]. LPS disrupts the stability of MMP, which is not conducive to maintaining normal physiological functions of cells [22,23,24]. The fluorescence of MMP and the fluorescence statistic data were shown in Figure 5A,B, respectively, LPS-induced increase of green fluorescence JC-1 was almost completely reversed to red fluorescence by DB, suggesting that DB restored LPS-induced MMP loss. 

Mitochondria are the main sites for ROS production. In this study, we found that the LPS-stimulation of macrophages resulted in ROS increases within the cells. On the contrary, DB decreased intracellular ROS level detected by the flow cytometry assay (Figure 5C,D), ROS kit (Figure 5E) and fluorescence assay (Appendix A), respectively.

### 2.6. DB Suppresses the LPS-Stimulated Inflammatory Response Through the Nrf2 Pathway

An oxidative stress pathway is an important mechanism of inflammatory response. Nrf2 has an important role in oxidative stress, which normally binds to its negative regulatory protein Keapl in the cytoplasm during a resting state [25]. 

When Nrf2 decouples from Keapl and transfer into the nucleus, it will combine with antioxidant response element (ARE) to initiate downstream antioxidant protein gene transcription [26]. Consequently, the expression of HO-1 inhibits the production of inflammatory mediators, while NQO1 constrains the production of oxygen free radicals during oxidative stress [14,27,28]. In our study, the results indicated that DB increased the mRNA expression of Nrf2 and HO-1 in RAW264.7 cells (Figure 6A,B). Besides, LPS significantly enhanced the expression of Keap-1, which was down-regulated by DB. DB pretreatment upregulated the protein expressions of Nrf2, HO-1, and NQO1 (Figure 6C). DB treatment decreased the expression of Nrf2 in the cytoplasm and increased the expression of Nrf2 in the nucleus respectively, meaning that DB pretreatment might promote Nrf2 translocation from the cytoplasm into the nucleus (Figure 6D). Furthermore, using confocal microscopy assay, the immunofluorescence analysis indicated that DB promoted Nrf2 translocation from cytoplasm into nucleus (Figure 6E).

## 3. Discussion

Inflammation is a defensive immune response to stimulation. Bacteria, viruses, endotoxins and so on can lead to inflammation of human tissues, organs and cells [29]. LPS is the main component of the cell wall of Gram-negative bacteria. Its toxic component is lipid A, which can induce an immune response of mammalian cells and lead to the release of pro-inflammatory factors. Macrophages are the main inflammatory cells in the process of inflammation and related diseases [30]. RAW264.7, a mouse macrophage, induced by LPS, is used as an inflammation model in vitro. LPS can induce cells to produce a variety of inflammatory mediators, cytokines, chemokines and similar, which in turn promote inflammation [31]. Therefore, this model is widely applied for drug screening and anti-inflammation evaluation. In this study, we employed RAW264.7 cells for the investigation of DB’s anti-inflammatory effect. Gao et al. illustrated the study of DB [18]. However, they mainly focused on the isolation and structure elucidation of DB and the part of anti-inflammatory activity. The mechanism need to be further studied. 

INOS, which is mainly expressed under the stimulation of inflammation and LPS [32], is closely related to inflammation. COX-2 is an inducible expression enzyme, whose expression is increased in various inflammatory reactions and is related to the severity of inflammation [33]. TNF-α, IL-1β, and IL-6 are pro-inflammatory cytokines and early inflammatory markers of the body produced by LPS-induced macrophages, which can be secreted and released in large quantities under conditions of injury, infection and immune response [34,35,36]. They are the triggers of inflammatory response. In this study, DB significantly suppressed LPS-induced nitrite level in RAW264.7 cells, which was furthermore confirmed by detecting iNOS expression using western blotting and NO generation using cytometry assay, respectively. In addition, TNF-α, IL-1β, and IL-6 cytokines release and gene expression were also inhibited by DB pretreatment with LPS-stimulated RAW264.7 cells. Collectively, DB displayed an obvious anti-inflammatory effect in vitro. Compared with the previous study [18], the dosage and the activity of DB are different. However, DB both displays significant anti-inflammatory activity in two different study. We speculate the main factors to lead to the small discrepancy. The experiments are performed, in two different laboratory, meaning many different experimental conditions, including the source of cells, machines and materials will lead to the small discrepancy. In addition, although DB is relatively stable, around 5 years have passed since we obtained the compound, so the DB may have degraded a little and lost part of its activity. Fortunately, so far, the purity of DB is over 98% (Appendix A), suggesting that DB is relatively stable. DB still has a significant anti-inflammatory activity, suggesting DB indeed is a potential lead-compound as anti-inflammatory agent for further research. 

The NF-κB signaling pathway is important in the pathological process of the inflammatory responses [37]. NF-κB is a key transcription factor in inflammatory responses. It exists in the cytoplasm in the form of inactive NF-κB/IκB complex at rest. When cells are stimulated by LPS, IκB is phosphorylated and degraded, and then NF-κB is transferred to the nucleus to regulate the expression of cytokines, chemokines and inflammatory response [38]. Our results suggested that DB significantly inhibited LPS-induced increased mRNA expression of NF-κB/p65 and IκBα and reversed the translocation of NF-κB from the cytoplasm into the nucleus in LPS-stimulated RAW264.7 cells. The results were confirmed by the immunofluorescence analysis and the cytoplasm and nucleus proteins separation kit. As a result, it appears that the NF-κB pathway participated in DB’s anti-inflammatory process.

Toll-like receptor 4 (TLR4) is the special pathogen pattern receptor on the cell surface, which can detect the exogenous homologous ligand LPS and activate the transcription factor NF-κB, leading to the expression of inflammatory cascade effector enzymes such as TNF-α, IL-1β, and IL-6. [39]. TLR4 can activate the downstream of myeloid differentiation factor 88 (MyD88)-dependent signaling pathway, including the PI3K/Akt and NF-κB pathways. Our results indicated that DB suppressed LPS-induced protein expression of TLR4and MyD88, and phosphorylation of TAK1, PI3K, and AKT, suggesting that DB exerts anti-inflammatory effects via TLR4/MyD88-PI3K/AKT pathway.

Reactive oxygen species (ROS) is a by-product of aerobic metabolism. It has strong chemical reactivity and an important role in cell signal transduction and body stability [40]. Once stimulated by LPS, ROS can sharply increase in RAW264.7 cells inducing inflammation through a series of signal transduction pathways via the oxidative stress pathway. In addition, excessive ROS can detriment mitochondria activity by decreasing MMP [41]. In this study, our data suggested that DB also exerts an anti-inflammatory effect by inhibiting ROS generation and MMP loss. 

Oxidative stress is caused by the imbalance between the production of reactive oxygen species and the body’s antioxidant defense, which have an important role in the inflammatory response [42,43]. Nuclear factor E2-related factor 2 (NRF2)/heme oxygenase 1 (HO-1) pathway is one of the most important antioxidant stress mechanisms *in vivo*, which participates in the antioxidant stress response of most tissues and organs [44]. Under normal physiological conditions, NRF2 is stored in the cytoplasm combined with the inhibitory protein Keap1 [45]. When stimulated by oxidative stress, NRF2 is dissociated from Keap1, transferred to the nucleus, and combined with antioxidant response elements (ARE), where it activates the transcription of the HO-1 gene to achieve antioxidant effects [46,47,48,49]. Our results showed that DB blocked LPS-induced translocation of Nrf2 from the nucleus into the cytoplasm, which was further confirmed by immunofluorescence analysis and the nuclear and cytoplasmic protein extraction kit. In conclusion, our data demonstrates that DB exhibits anti-inflammatory and anti-oxidant effects in LPS-stimulated RAW264.7 cells via the TLR4/MyD88-PI3K/AKT NF-κB, and Nrf2/HO-1 signaling pathways.

## 4. Materials and Methods

### 4.1. Reagents and Chemicals

The purity of DB (over 98%) isolated from Danshen in our laboratory was determined by high-performance liquid chromatography (HPLC). LPS (Lipopolysaccharides from *Escherichia coli* O111:B4), Griess reagent (modified-G4410), 2′,7′-dichlorodihydrofluorescein diacetate (DCFH2-DA), methylthiazolyldiphenyl-tetrazolium bromide (MTT), 5,5′,6,6′-tetrachloro-1,1′,3,3′-tetraethyl-benzimidazolylcarbocyanine iodide (JC-1), MAK-143 intracellular ROS kit and dimethyl sulfoxide (DMSO) were purchased from Sigma-Aldrich (St. Louis, MO, USA). Dulbecco’s modified eagle medium (DMEM), NO detector DAF-FM, fetal bovine serum (FBS) were acquired from Life Technologies/Gibco Laboratories (Grand Island, NY, USA). IL-6 and TNF-α ELISA kits were obtained from Neonbioscience (Shenzhen, China). Antibodies against iNOS (#13120), COX-2 (#4842), NF-κB/p65 (#8242T), Keap-1 (#4678), Nrf2 (#12721), HO-1 (#70081), NQO1,PARP (#9532), TAK1 (#5206), p-TAK1 (#9939), MyD88 (#4283), TLR4 (#14358), p-PI3K (#4228), PI3K (#4249), Akt (#4691), p-Akt (#4060), mTOR (#2983), p-mTOR (#5536),and GAPDH (#5174) were obtained from Cell Signaling (Beverly, MA, USA). Nuclear and Cytoplasmic Protein Extraction kit was obtained from Beyotime (Shanghai, China). 

### 4.2. Qualtiy Control of DB

The purity of DB was determined by HPLC (Accutiy/Arc with 2998, Waters, Milford, MA, USA). An ODS column (CORTECS^®^ C18, 2.7µm, 4.6 × 50 mm) was used. The following mobile phase: acetonitrile/water (50/50) with a flow rate of 1.0 mL/min was adopted. The absorbance wavelength of the detector was set at 220 and 430 nm.

### 4.3. Cell Culture

RAW264.7 macrophages cells were purchased from the Cell Bank of the Chinese Academy of Sciences (Shanghai, China). Cells were cultured in DMEM supplemented with 10% FBS in a humidified atmosphere containing 5%CO_2_/95% air at 37 °C.

### 4.4. MTT Assay

Cells were cultured in 96-well plates at a density of 5 × 104 per well overnight and then treated with DB (0, 5, 10, and 20 µM). After 24 h, MTT dye (5 mg/mL) was added to each well and incubated for another 3 h at 37 °C. After removal of the medium, 100 µL of DMSO was added to each well and properly mixed for another 10 min. The absorbance at 570 nm was determined using a microplate reader.

### 4.5. Griess Reagent Assay

RAW264.7 cells were seeded in 12-well plates at a density of 1 × 10^5^ per well and cultured overnight. Cells were first pretreated with DB (0, 5, 10, and 20 µM) for 1 h, and then incubated in 1 µg/mL LPS for another 24 h. The medium was collected to determine the nitrite level using the Griess assay.

### 4.6. Assessment of Cytokine Release

TNF-α and IL-6 were investigated by ELISA following the manufacturer’s instructions. Cells were plated into 24-well plates overnight. Consequently, cells were pretreated with DB (0, 5, 10, and 20 µM) for 1 h, following incubation in DB and LPS (1 µg/mL) for 24 h. The medium was determined for the determination of the release of the cytokine.

### 4.7. ROS Kit

RAW264.7 cells were cultured in 96-well black plates with a clear bottom at a density of 5 × 10^4^ per well overnight. Cells were then pretreated with DB (0, 5, 10, and 20 µM) for 1 h. Consequently, cells were cultured with or without LPS (1 µg/mL) for another 8 h. Then, MAK-143 intracellular ROS kit (Sigma-AldrichLouis, MO, USA) was used to measure the level of intracellular ROS according to the manufacturer’s instruction. In a brief, add 100 µL/well of Master Reaction Mix into cell plate and incubate for 1 h in an incubator. Then the fluorescence intensity was determined by a fluorescence microplate reader (λex = 490/λem = 525 nm).

### 4.8. Flow Cytometry Assay

RAW264.7 cells were seeded in 12-well plates at a density of 2 × 10^5^ cells per well and cultured overnight. Subsequently, cells were pretreated with DB (0, 5, 10, and 20 µM) for 1 h, following treatment with or without LPS (1 µg/mL). Different probes were used to detect the corresponding indicators, including NO detector DAF-FM (1 µM, 1 h) and ROS detector DCFH2-DA (1 µM, 30 min). After probes incubation, cells were collected and tested by flow cytometry (Becton-Dickinson, Franklin Lakes, NJ, USA).

### 4.9. Western Blotting Analysis

RAW264.7 cells were seeded in a dish or 6-well plates and fostered overnight. The next day, cells were treated with DB (0, 5, 10, and 20 µM) and LPS for a certain period of time. Total cell proteins were extracted using RIPA (1% PMSF and 1% cocktail). The cytoplasmic extraction and nuclear protein kit (Beyontime, Shanghai, China) were used to obtain the cytoplasmic and nuclear proteins, following the manufacturer’s protocol. A BCA protein kit (Thermofisher, Waltham, MA, USA) was employed to determine protein concentrations. The denatured proteins were then separated by 8% or 10% SDS-PAGE gels and transferred to PVDF membrane (Millipore, Billerica, MA, USA). After blocking the PVDF membrane with 5% nonfat milk for 1h, the PVDF membrane was incubated with primary antibodies (1:1000) at 4 °C for more than 12 h. After wash with TBST and incubation with secondary antibody (1:5000) for 2 h at room temperature, the membranes were analyzed with ChemiDoc™ MP Imaging System (Bio-Rad, Hercules, CA, USA). GAPDH was used as a housekeeping protein.

### 4.10. Quantitative Real-Time PCR (qRT-PCR) Assay

RAW264.7 cells were seeded at a density of 2 × 10^6^ cells per well into a dish and cultured overnight. Cells were pretreated with DHT (20 µM) for 1 h, following treatment with LPS (1 µg/mL) for another 2 h. Total RNA was extracted using a Trizol assay kit. Subsequently, RNA (1 µg) was subjected to qRT-PCR using a qPCR master mix kit (Applied Biosystems, PowerSYBR Green PCR Master Mix, 4367659, London, UK). PCR amplification was performed by incorporating SYBR green. The oligonucleotide primers for mouse IL-1β, IL-6, TNF-α, p65, IκBα, HO-1, NRF2, and GAPDH were synthesized by Invitrogen (Nanning, China); the gene sequences were as follows:

IL-1β-FGAAAGACGG CACACCCACCCTIL-1β-RGCTCTGCTTGTGAGGTGCTGATGTAIL-6-FTCCAGTTGCCTTC TTGGGACIL-6-RGTGTAATTAAGCCTCCGACTTGTNF-α-FTTCTGTCTACTGAACTTCGGGGTGATCGGTCCTNF-α-RGTATGAGATAGCAAATCGGCTGACGGTGTGGGNF-κB/p65-FGCACGGATGACAGAGGCGTGTATAAGGNF-κB/p65-RGGCGGATGATCTCCTTCTCTCTGTCTGIκB-α-FTGCTGAGGCACTTCTGAGIκB-α-RCTGTATCCGGGTGCTTGGHO-1-FTCAGTCCCAAACCTCGCGGTHO-1-RGCTGTGCAGGTGTTAGCCNRF2-FAGCAGGACATGGAGCAAGTTNRF2-RTTCTTTTTCCAGCGAGGAGAGAPDH-FCATGACCACAGTCCATGCCATCACGAPDH-RTGAGGTCCACCACCC TGTTGCTGT

### 4.11. Immunofluorescence Assay

RAW264.7 cells were seeded in confocal dishes (SPL, Pocheon, Korea) with a density of 20 × 10^5^ cells per well overnight. Cells were then pretreated with DB (20 µM) for 1 h and LPS-stimulated (1 µg/mL) for 1 h. Consequently, cells were fixed, punched, blocked, and then incubated with NF-κB p65 or NRF2 antibody (1:100) overnight at 4 °C. Finally, cells were incubated with Alexa Fluor 594 secondary antibody for 1 h. Nuclei were revealed by Hoechst 33342 staining. Fluorescence images were collected under a confocal microscope system (Leica, Wetzlar, Germany).

### 4.12. Fluorescence Assay

RAW264.7 cells were seeded in 96-well plates with a density of 2 × 10^5^ cells per well overnight. Cells were pretreated with DB (20 µM) for 1h and then incubated with or without LPS for another 8 h. Consequently, cells were stained with JC-1 (10 µg/mL) and DCFH_2_-DA (100 µM) for 30 min. Fluorescence images were captured using a fluorescence microscopy (Leica, DMi8, Wetzlar, Germany).

### 4.13. Statistical Analysis

Data are presented as means ±SD. All experiments were repeated at least three times. Data were normally distributed and analyzed by *one-way-ANOVA* by Graph Pad Prism 7 software (Microsoft, Seattle, WA, USA). A *P* value < 0.05 was considered to be statistically significant.

## Figures and Tables

**Figure 1 molecules-24-04502-f001:**
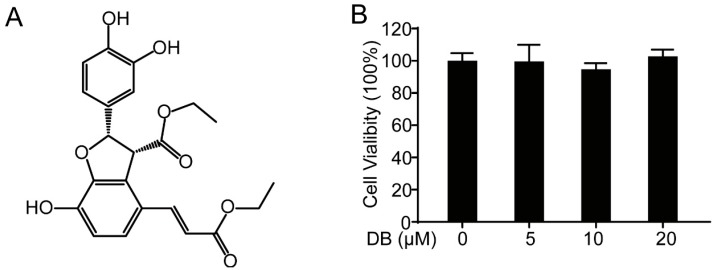
DB suppressed the inflammatory response in LPS-stimulated RAW264.7 cells. (**A**) The chemical structure of DB. (**B**) The cytotoxicity of DB (0, 5, 10, and 20 µM) analyzed by MTT assay after 24 h treatment, (n = 5). (**C**) RAW264.7 cells were pretreated with DB (0, 5, 10, and 20 µM) for 1 h and then stimulated with LPS (1 µg/mL) for 18 h. The levels of nitrite were determined by the Griess assay, (n = 5). (**D**) Cells were pretreated with DB (0, 5, 10, and 20 µM) for 1 h and co-cultured with LPS (1 µg/mL) for another 18 h. Total proteins were collected and the expressions of iNOS and COX-2 were detected by Western blotting, (n = 3). (**E**) Cells were stimulated with LPS (1 µg/mL) for 8 h with or without DB (20 µM) pretreatment for 1 h; the NO levels were measured by flow cytometry, (n = 3). (**F**) Statistical analysis of the NO per group. ** *p* < 0.01, and *** *p* < 0.001 vs. LPS group.

**Figure 2 molecules-24-04502-f002:**
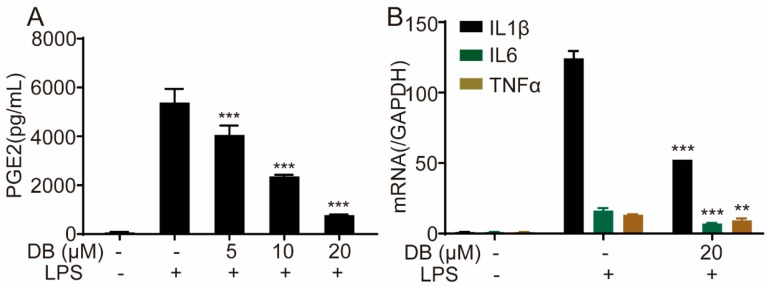
DB suppressed the release and gene expression of pro-inflammatory cytokines PGE2 in LPS-induced RAW264.7 cells. RAW 264.7 cells were pretreated with DB for 1 h and then stimulated with LPS (1 µg/mL) for 24 h. The levels of PGE2 (**A**) in the culture medium were determined by ELISA kits. RAW264.7 cells pretreated with DB (20 µM) for 1 h were stimulated with LPS (1 µg/mL) for 6 h, after which IL-6, TNF-α, and IL-1β mRNA levels were determined by qRT-PCR, (**B**) mRNA levels were determined by qRT-PCR, (n = 3). ** *p* < 0.01, and *** *p* < 0.001vs LPS group.

**Figure 3 molecules-24-04502-f003:**
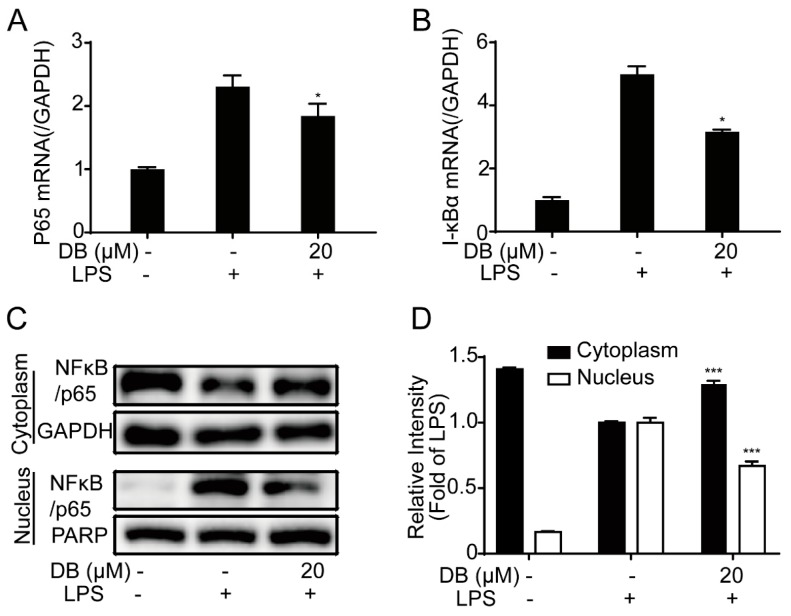
DB suppressed the LPS-induced NF-κB nuclear translocation in RAW264.7 cells. RAW264.7 cells pretreated with DB (20 µM) for 1 h were stimulated with LPS (1 µg/mL) for 6 h, after which NF-κB/p65 (A) and IκBα (B) mRNA levels were determined by qRT-PCR. RAW264.7 cells were pretreated with DB (20 µM) for 1 h and then stimulated with LPS (1 µg/mL) for 2 h, (n = 3). The protein expression of p65 in cytoplasm and nucleus was detected by western blotting, (n = 3) (C,D). The localization of p65 in the cytoplasm and nuclear was measured by immunofluorescence staining. Seven to nine cells were chosen for observation, (n = 3) (E). * *p* < 0.05, and *** *p* < 0.001vs LPS group.

**Figure 4 molecules-24-04502-f004:**
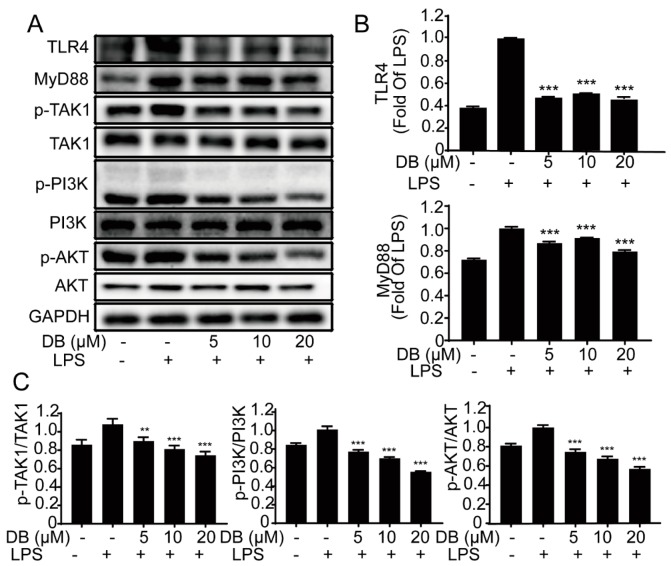
DB suppressed the LPS-stimulated TLR4-MyD88 pathway. RAW264.7 cells were stimulated with LPS (1 µg/mL) for 2 h to 8 h with or without DB (20 µM) pretreatment for 1 h. The protein expression of TLR4, MyD88, and phosphorylation of TAK1, PI3K, AKT were measured by western blotting, (n = 3) (**A**). Statistical analyses were performed for each group of TLR4, MyD88 (**B**) and p-TAK1, p-PI3K, p-AKT (**C**). n = 3. ** *p* < 0.01, and *** *p* < 0.001 vs. LPS group.

**Figure 5 molecules-24-04502-f005:**
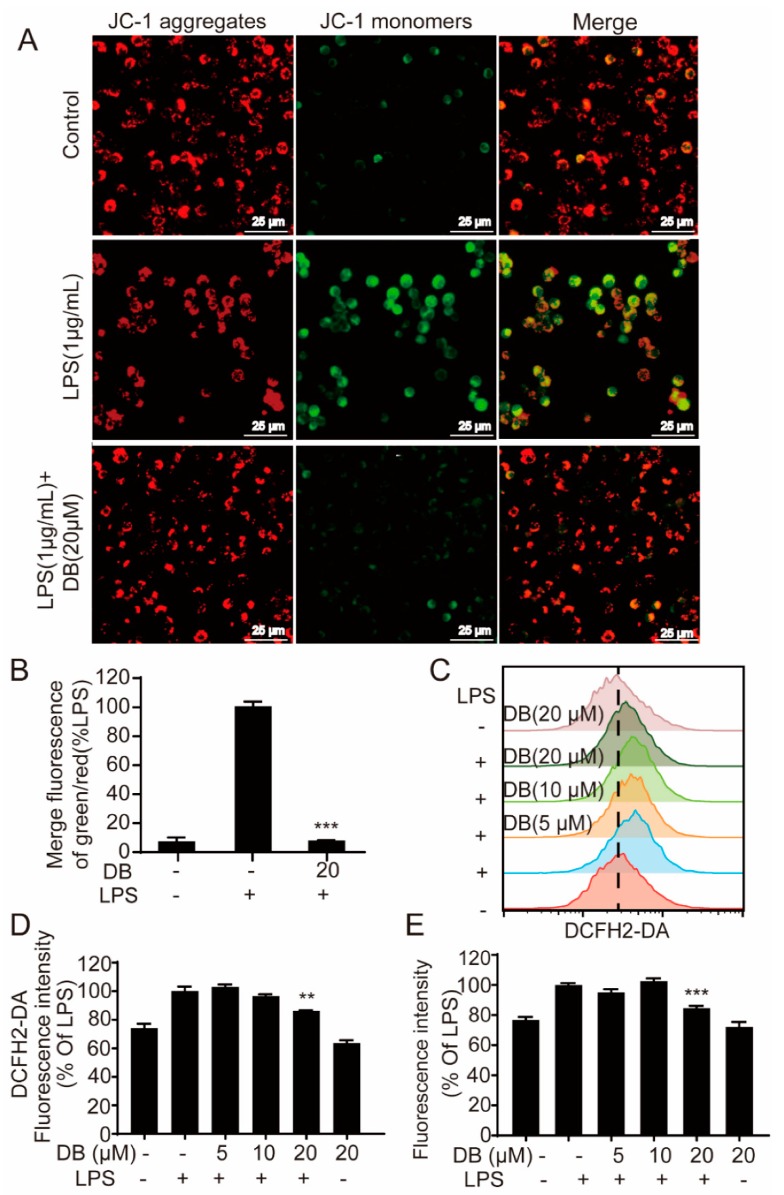
DB suppressed the loss of MMP and ROS generation. (**A**) Cells were stimulated with LPS (1 µg/mL) for 8 h with or without DB (20 µM) pretreatment for 1 h, the MMP was detected with JC-1 and the images were captured by fluorescence microscopy (Leica DMi8), (n = 3). (**B**) The statistical analysis of the ratio of green/red fluorescence for the corresponding images. (**C**) Cells pretreated with indicated concentrations of DB for 1 h were co-cultured with LPS (1 µg/mL) for another 6 h. Then cells labeled with DCFH_2_-DA (1 µM) for 30 min were investigated by flow cytometry, (n = 3). (**D**) The statistical analysis of fluorescence intensity of the flow cytometry results. (**E**) Cells pretreated with indicated concentrations of DB for 1 h were co-cultured with LPS (1 µg/mL) for another 6 h. Add 100 µL/well of Master Reaction Mix into a cell plate and incubate for 1 h in an incubator. Then the fluorescence intensity was determined by a fluorescence microplate reader (λex = 490/λem = 525 nm), (n = 4).

**Figure 6 molecules-24-04502-f006:**
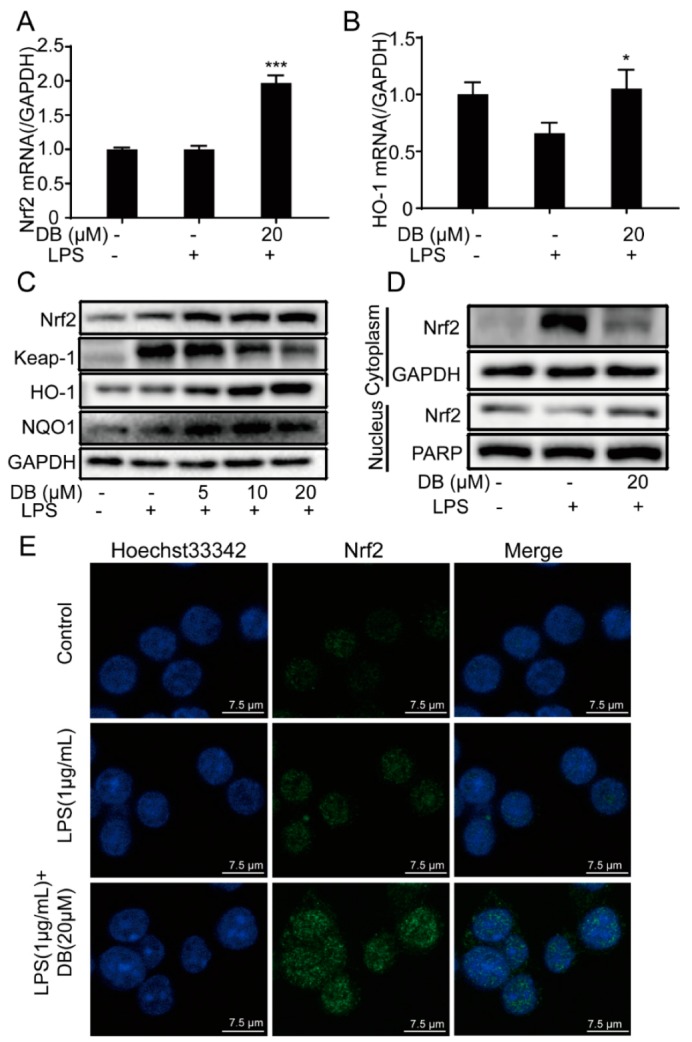
DB suppressed LPS-stimulated inflammatory response through Nrf2 pathways. RAW264.7 cells pretreated with DB (20 µM) for 1 h were stimulated with LPS (1 µg/mL) for 6 h, after which Nrf2 (**A**) and HO-1 (**B**) mRNA levels were determined by qRT-PCR, (n = 3). (**C**) RAW 264.7 cells were stimulated with LPS (1 µg/mL) for 18 h with or without DB (0, 5, 10, and 20 µM) pretreatment for 1 h. The protein expression of Nrf2, Keap1 HO-1, and NQO1 were measured by western blotting, (n = 3). (**D**) The protein expressions of Nrf2 in cytoplasm and nucleus were detected by western blotting after induced by LPS for 2 h with or without DB (20 µM) pretreatment, (n = 3). (**E**) The localization of Nrf2 in the cytoplasm and nucleus was measured by immunofluorescence staining, (n = 3). * *p* < 0.05, and *** *p* < 0.001 VS. LPS group.

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
