# Peer review of "Diethyl Blechnic Exhibits Anti-Inflammatory and Antioxidative Activity via the TLR4/MyD88 Signaling Pathway in LPS-Stimulated RAW264.7 Cells"

_molecules, 2019, doi:10.3390/molecules24244502_

Round 1

Reviewer 1 Report

One main reason for the previous rejection of the manuscript was that the authors have previously published very similar results (Gao et al, Tanshinones and diethyl blechnis with anti-inflammatory and anticancer activities from Salvia miltiorrhiza Bunge; Sci Rep. 2016 and another paper in Sci Rep. In 2018). Unfortunately, the authors have only once (chapter 2.1) added a sentence that their previous results have been confirmed. Mostly they marginally changed the wording in the resubmitted manuscript. It is urgently important that the authors can mention the known results in the manuscript (may be for a better understanding), but they must remove the known results from the manuscript and put them in Supporting Information.

One important point came out: the questionable purity of the used compound DB! This must be clarified (see the comment in the discussion part).

The authors have a little bit improved their manuscript. However, improvements are still urgently necessary. Moreover, some critical points from the last review have not been tackled.

In detail:

Abstract:

The abstract should only contain the new results. In the present version the authors have only removed some words, such as „furthermore“ or „also“.

Results:

2.2.: Il-6 and TNF-alpha secretion have already been reported. Therefore the results can shortly be mentioned, but the respective figure should be put in Supporting Information.

2.4.: The authors have replaced Fig. 4B. What was the reason? Moreover, they did not mention the missing concentration dependency.

2.5.: Fig. 5 legend is still not sufficient and confusing. It should be made clearer that Fig. 5B is related to Fig. 5A. It is not clear that Fig. 5C and Fig. 5D belong together. It is not sufficient to write for Fig. 5D: „The statistical analysis of fluorescence intensity“. In Fig 5D ROS should be replaced by DCFH2-DA. It should be also mentioned that ROS was measured using the DCFH2-DA assay, once by flow cytometry, and once in the fluorescence microplate reader.

2.6.: Fig. 6 legend is still confusing. What is shown in Fig. 6E? Nrf2 in the cytoplams? The authors wrote: „“…Nrf2 in the cytoplasm and nuclear“ (should be at least nucleus

Looking at Fig. 6B there is no „dramatical increase of HO-1 mRNA compared to the control. Results shown in Fig. 6A and B can only be analyzed if the cells are also treated with DB alone, as in Fig. 6A LPS has no effect, but the combination of DB and LPS. In Fig. 6B LPS decreases HO-1 mRNA, but the combination of DB(LPS brings it to the control level. Which treatemnt is reasonable for the respective effect? These experments have tob e performed.

Results from HO-1 mRNA in Fig. 6B do not fit to those ones from Ho-1 secretion in Fig. 6C.

From their results they cannot conclude that Nrf2 expression is influenced by DB.They have only detected the protein itself. As they have not performed Nrf2 location in the cytoplasm and in the nucleus in the same Western blot, nothing can be concluded in a scientifically sound manner.

Discussion:

The authors mentioned the study of Gao et al, bt they wrote „The mechanism remained to to be unraveled“. This is wrong, as the paper from Gao et al also elucidated a part of the mechanism.

All chapters containg previous published results can be shortened, e.g. the chapter on page 19 on iNOS.

The anti-inflammatory activity can only be proven in in vivo experiments, but not in cell-based assays. Please correct the wording on page 19, line 241.

Page 21, line 267: DB cannot suppress expression of p-TAK1, pPI3K and pAKT, but inhibit the phosphorylation of the respective protein.

One very important point: The authors wrote as reason for the discrepancies that DB may be degraded in the last five years (lines 245 – 246). The purity of the studied compound is essentiell. Therefore, the authors must check the purity grade and include it in the manuscript! In case that DB is degraded the results are questionable.

Author Response

Response to Reviewer 1 Comments

Point 1: One main reason for the previous rejection of the manuscript was that the authors have previously published very similar results (Gao et al, Tanshinones and diethyl blechnis with anti-inflammatory and anticancer activities from Salvia miltiorrhiza Bunge; Sci Rep. 2016 and another paper in Sci Rep. In 2018). Unfortunately, the authors have only once (chapter 2.1) added a sentence that their previous results have been confirmed. Mostly they marginally changed the wording in the resubmitted manuscript. It is urgently important that the authors can mention the known results in the manuscript (may be for a better understanding), but they must remove the known results from the manuscript and put them in Supporting Information.

One important point came out: the questionable purity of the used compound DB! This must be clarified (see the comment in the discussion part).

The authors have a little bit improved their manuscript. However, improvements are still urgently necessary. Moreover, some critical points from the last review have not been tackled.

Response 1: We greatly appreciate your professional comments. Your comments from the editors and reviewers have been addressed point by point as the follows.

Point 2: Abstract:

The abstract should only contain the new results. In the present version the authors have only removed some words, such as „furthermore“ or „also“.

 Response 2: We have revised the Abstract section. The known results have been removed.

Results:

Point 3: Il-6 and TNF-alpha secretion have already been reported. Therefore, the results can shortly be mentioned, but the respective figure should be put in Supporting Information.

Response 3: Thanks for your kind reminder. IL-6 and TNF-alpha secretion results have been removed, which were put in the supporting information.

Point 4: The authors have replaced Fig. 4B. What was the reason? Moreover, they did not mention the missing concentration dependency.

Response 4: The previous statistical analysis is not consistent with western blotting results. Thus, we have corrected and replaced results of the previous statistical analysis. For the TLR4 western blotting results, we have repeated at least 3 times. Unfornately, DB did not exhibit an inhibitory effect in a concentration-dependent manner.

Point 5: Fig. 5 legend is still not sufficient and confusing. It should be made clearer that Fig. 5B is related to Fig. 5A. It is not clear that Fig. 5C and Fig. 5D belong together. It is not sufficient to write for Fig. 5D: „The statistical analysis of fluorescence intensity“. In Fig 5D ROS should be replaced by DCFH2-DA. It should be also mentioned that ROS was measured using the DCFH2-DA assay, once by flow cytometry, and once in the fluorescence microplate reader.

Response 5: Thanks for your kind reminder. We have added the explanation in figure 5 legend. Fig.5B is the statistical analysis in terms of the images of Fig.5A. For Fig. 5C and 5D, we have corrected figure as the follows:

Point 6: legend is still confusing. What is shown in Fig. 6E? Nrf2 in the cytoplams? The authors wrote: „“…Nrf2 in the cytoplasm and nuclear“ (should be at least nucleus

Looking at Fig. 6B there is no „dramatical increase of HO-1 mRNA compared to the control. Results shown in Fig. 6A and B can only be analyzed if the cells are also treated with DB alone, as in Fig. 6A LPS has no effect, but the combination of DB and LPS. In Fig. 6B LPS decreases HO-1 mRNA, but the combination of DB(LPS brings it to the control level. Which treatemnt is reasonable for the respective effect? These experments have to be performed.

Results from HO-1 mRNA in Fig. 6B do not fit to those ones from Ho-1 secretion in Fig. 6C.

From their results they cannot conclude that Nrf2 expression is influenced by DB.They have only detected the protein itself. As they have not performed Nrf2 location in the cytoplasm and in the nucleus in the same Western blot, nothing can be concluded in a scientifically sound manner.

Response: Thanks for your kind reminder. We have replaced the nucleus with nuclear in the legend of Fig 6E; In the previous study showed, the level of Nrf2 and HO-1are not stable after treatment with LPS, but if the compounds or drugs has anti-inflammatory effect, the expression of Nrf2 or HO-1 will be increased; The mRNA level of HO-1 has been detected at 6h, which is an early stage of DB exerted anti-inflammatory effect, so it is not dramatically compared with control group, but there is an increasing compared with LPS treatment; As the times are different in the determination of mRNA(6h) and the protein expression (18h) after treatment with LPS, so the result in Fig 6C are not consistent with Fig 6B; In Fig 6D and Fig 6E, the location of Nrf2 were detected by western blotting and immunofluorescence respectively, the results showed that DB increased the expression of Nrf2 in nucleus. So we conclusion that DB has effect on Nrf2 expression was influenced by DB.

Discussion:

The authors mentioned the study of Gao et al, bt they wrote „The mechanism remained to to be unraveled“. This is wrong, as the paper from Gao et al also elucidated a part of the mechanism.

All chapters containg previous published results can be shortened, e.g. the chapter on page 19 on iNOS.

The anti-inflammatory activity can only be proven in in vivo experiments, but not in cell-based assays. Please correct the wording on page 19, line 241.

Page 21, line 267: DB cannot suppress expression of p-TAK1, pPI3K and pAKT, but inhibit the phosphorylation of the respective protein.

Response 7: We have

All chapters containg previous published results can be shortened, e.g. the chapter on page 19 on iNOS.

Response 8:

The anti-inflammatory activity can only be proven in in vivo experiments, but not in cell-based assays. Please correct the wording on page 19, line 241.

Response 9:

Page 21, line 267: DB cannot suppress expression of p-TAK1, pPI3K and pAKT, but inhibit the phosphorylation of the respective protein.

Response 10: Thanks for your kind reminder. We have revised in the manuscript as your comments.

Point : One very important point: The authors wrote as reason for the discrepancies that DB may be degraded in the last five years (lines 245 – 246). The purity of the studied compound is essentiell. Therefore, the authors must check the purity grade and include it in the manuscript! In case that DB is degraded the results are questionable.

Response: Thanks for your kind reminder. We have detected the purity of DB by HPLC at the conditions of the follows: the mobile flow phase: acetonitrile/water (50/50), the flow rate: 1 mL/min, the wave length: 430 and 220 nm. We have a conclusion that the purity of DB is over 95%, suggesting that DB is relatively stable (Fig S3).

Reviewer 2 Report

This is a well designed research project and a MS that merits publication. I would like to make only one minor suggestion to the authors:

inflammation is linked to a number of diseases,

I would encourage the authors to include these two papers in their introduction:

https://www.mdpi.com/2072-6643/10/5/604 and https://www.mdpi.com/2072-6643/11/10/2332

This will make their introduction stronger and more to the point that it is of vital importance to inhibit inflammation.

I would suggest minor revision.

Author Response

This is a well designed research project and a MS that merits publication. I would like to make only one minor suggestion to the authors:

inflammation is linked to a number of diseases,

I would encourage the authors to include these two papers in their introduction:

https://www.mdpi.com/2072-6643/10/5/604 and https://www.mdpi.com/2072-6643/11/10/2332

This will make their introduction stronger and more to the point that it is of vital importance to inhibit inflammation.

I would suggest minor revision.

Response: Thanks for your kind reminder. We have citation these two papers in the manuscript.

Reviewer 3 Report

Authors described that Diethyl Blechnic exhibits anti-inflammatory and antioxidative 1 activity via the TLR4/MyD88 Signaling Pathway in LPS-2 stimulated RAW264.7 Cells.

The experimental results are interesting but need some supplementation.

Figure 1E and F results are not significant, so it's better to measure NO production by Griess assay. Figure 2, C, D and E should be indicated electophoresis results with histogram. Figure 3A and B also should be indicated electrophoresis results with histogram. Figure D and E,ROS and NO production are not significant even at high concentrations of LPS, so it's necessrary to show different time and measurement methods. PCR primers should be arranged in a table in Materials and Methods

Author Response

Comments and Suggestions for Authors

Authors described that Diethyl Blechnic exhibits anti-inflammatory and antioxidative 1 activity via the TLR4/MyD88 Signaling Pathway in LPS-2 stimulated RAW264.7 Cells.

The experimental results are interesting but need some supplementation.

Figure 1E and F results are not significant, so it's better to measure NO production by Griess assay. Figure 2, C, D and E should be indicated electophoresis results with histogram. Figure 3A and B also should be indicated electrophoresis results with histogram. Figure D and E,ROS and NO production are not significant even at high concentrations of LPS, so it's necessrary to show different time and measurement methods. PCR primers should be arranged in a table in Materials and Methods 

Response: Thanks for your kind reminder. In the manuscript, Figure 1C are Griess assy result of NO production. And we put Figure 2 in supplement information and the main results is about PGE2 level in Figure 2A. Figure 3A and B are mRNA level histogram. Figure 3D are the histogram of Figure3B. And the PCR primers are arranged in a table in Materials and Methods as follow:

Reviewer 4 Report

In this paper, He et al. analyze the anti-inflammatory effect of Diethyl Blechnic (DB). More specifically they study how DB impacts on the TLR4 pathway after exposure of Raw264.7 cells to LPS.

This work complements what was previously published by the same labs (Gao et al. 2016) and a large part of the study, related to NF-kB, does not bring many new data. For instance, several analysis presented in Fig 1 (Panels C,D and E), Fig 2 (Panels A and B) and Fig 3 (Panel E) were already reported in Gao et al. publication with compound #4 being DB. New data concern analysis at other levels of the same pathway. It would have been more interesting to assess issues related to the molecular target of DB in the LPS/NF-kB pathway rather than repeating experiments. Previously, dimerization of the LPS receptor, TLR4, was claimed to be affected by DB. Is it indeed the primary target in the LPS/TLR signaling pathway and does this result in a stimulus-specific action of DB? This putatively important finding would be strengthened by showing what happens with DB when TNF is used as an NF-kB activator instead of LPS.   

DB is also proposed to act at the level of Nrf2/Keap1 to modulate the oxidative stress response. This would represent an interesting new finding. Nevertheless, the data are very difficult to interpret since chaotically described in the result section. For instance, it is claimed that upon LPS treatment Nrf2 translocates from the nucleus to the cytoplasm, something that is the opposite of what is expected. Indeed, the authors show that LPS increases the ROS (said lines 162/163), something that should result in Nrf2 translocation into the nucleus (as explained line 183/184).

Since, at the same time, LPS modulates the expression of Nrf2, a more likely explanation would be that the observed increase of Nrf2 in the cytoplasm results from this induced synthesis rather than from a translocation from the nucleus. This makes the observed effect of DB difficult to connect to a true modulation of the oxidative response by Nrf2/Keap1. Last, but not least, examination of Fig. 6E do not show any cytoplasmic increase of Nrf2 upon LPS stimulation. Instead, this is observed upon LPS/DB treatment and is the opposite of what is said in the text.

All these inconsistencies seriously damage the only putatively innovative part of this work.

Author Response

Response to Reviewer 4 Comments

Comments and Suggestions for Authors

In this paper, He et al. analyze the anti-inflammatory effect of Diethyl Blechnic (DB). More specifically they study how DB impacts on the TLR4 pathway after exposure of Raw264.7 cells to LPS.

This work complements what was previously published by the same labs (Gao et al. 2016) and a large part of the study, related to NF-kB, does not bring many new data. For instance, several analysis presented in Fig 1 (Panels C,D and E), Fig 2 (Panels A and B) and Fig 3 (Panel E) were already reported in Gao et al. publication with compound #4 being DB. New data concern analysis at other levels of the same pathway. It would have been more interesting to assess issues related to the molecular target of DB in the LPS/NF-kB pathway rather than repeating experiments. Previously, dimerization of the LPS receptor, TLR4, was claimed to be affected by DB. Is it indeed the primary target in the LPS/TLR signaling pathway and does this result in a stimulus-specific action of DB? This putatively important finding would be strengthened by showing what happens with DB when TNF is used as an NF-kB activator instead of LPS.   

DB is also proposed to act at the level of Nrf2/Keap1 to modulate the oxidative stress response. This would represent an interesting new finding. Nevertheless, the data are very difficult to interpret since chaotically described in the result section. For instance, it is claimed that upon LPS treatment Nrf2 translocates from the nucleus to the cytoplasm, something that is the opposite of what is expected. Indeed, the authors show that LPS increases the ROS (said lines 162/163), something that should result in Nrf2 translocation into the nucleus (as explained line 183/184).

Since, at the same time, LPS modulates the expression of Nrf2, a more likely explanation would be that the observed increase of Nrf2 in the cytoplasm results from this induced synthesis rather than from a translocation from the nucleus. This makes the observed effect of DB difficult to connect to a true modulation of the oxidative response by Nrf2/Keap1. Last, but not least, examination of Fig. 6E do not show any cytoplasmic increase of Nrf2 upon LPS stimulation. Instead, this is observed upon LPS/DB treatment and is the opposite of what is said in the text.

All these inconsistencies seriously damage the only putatively innovative part of this work.

Response: Thanks for your kind reminder. We will further study the new targets of DB on LPS induced inflammatory as your suggestion. DB has effect on the LPS/TLR4 pathway, the results are showed in Figure 4A. The oxidative response of Nrf2 upon DB treatment after LPS stimulation, DB will increase the expression of Nrf2 in nucleus. The Nrf2 protein expression determined by western blotting and immunofluorescence as shown in Fig 6 C and E, and the increased expression of Nrf2 was in nucleus as shown in Figure 6 D. Nrf2 has been found to be a highly unstable protein (t1⁄2  15 min). Nrf2 activity is regulated in part by the actin-associated Keap1 protein, which was proposed to act by binding and tethering the transcription factor in the cytoplasm. Once activated in response to stress signals, this association will be, releasing Nrf2 for translocation into the nucleus to effect its transcriptional activity. In non-stress cells, Nrf2 binds to Keap1, which can inhibit Nrf2 activity and promote the ubiquitination of Nrf2. When oxidative stress occurs, Nrf2 will decoupled from Keap1 to avoid the degradation of ubiquitin and return to a highly unstable state. After Nrf2 enters the nucleus, it combines with ARE to activate the downstream protein to have antioxidant effect. Based on our experimental results, we speculate that DB can promote the entry of Nrf2, keep its activation stable, increase its expression and further promote the antioxidant effect.

Round 2

Reviewer 1 Report

Some minor corrections are necessary

2.5.: line 139: „..fuluorescence of MMP and the fluorescence statistic data were showed, please correct the spelling...“

2.6.: The response of the authors is not very convincing. Still they used the wordíng in line 165: „DB dramatically increased the 165 mRNA expression of Nrf2 and HO-1 in RAW264.7 cells“. Please delete the word dramatically.

Purity control: Please put the purity control in the manuscript!

Author Response

2.5.: line 139: „..fuluorescence of MMP and the fluorescence statistic data were showed, please correct the spelling...“

Reply: Thanks for your kind reminder. We have corrected the spelling error. Fuluoresecence----fluorescence; showed ----shown.

2.6.: The response of the authors is not very convincing. Still they used the wordíng in line 165: „DB dramatically increased the 165 mRNA expression of Nrf2 and HO-1 in RAW264.7 cells“. Please delete the word dramatically.

Reply: Thanks for your professional comments. The word "dramatically" was deleted. 

Purity control: Please put the purity control in the manuscript!

Reply: Thanks for your kind reminder. The assay and results of the quality control of DB were inserted in our manuscript. 

Reviewer 4 Report

N/A

Author Response

Reply: Thanks for your professional comments. The English language has been revised.